



# An Improved Representation of Fire Non-Methane Organic Gases (NMOGs) in Models: Emissions to Reactivity

Therese S. Carter[1], Colette L. Heald[1,2], Jesse H. Kroll[1], Eric C. Apel[3], Donald Blake[4], Matthew Coggon[5], Achim Edtbauer[6], Georgios Gkatzelis[5,7,#], Rebecca S. Hornbrook[3], Jeff Peischl[5,7], Eva Y. Pfannerstill[6*], Felix Piel[8.9], Nina G. Reijrink[6,10], Akima Ringsdorf[6], Carsten Warneke[5], Jonathan Williams[6], Armin Wisthaler[9,11], and Lu Xu[12,**]

[1]Civil and Environmental Engineering Department, Massachusetts Institute of Technology, Cambridge, MA 02139, USA

[2]Earth, Atmospheric and Planetary Sciences, Massachusetts Institute of Technology, Cambridge, MA 02139, USA

[3]Atmospheric Chemistry Observations & Modeling Laboratory, National Center for Atmospheric Research, Boulder, CO 80301, USA

[4]Chemistry Department, University of California Irvine 92697

[5]NOAA Chemical Sciences Laboratory, Boulder, CO, 80305, USA

[6]Atmospheric Chemistry Department, Max Planck Institute for Chemistry, 55128, Mainz, Germany

[7]Cooperative Institute for Research in Environmental Sciences, University of Colorado Boulder, Boulder, CO, USA

[8]Ionicon Analytik, Innsbruck, Austria

[9]Department of Chemistry, University of Oslo, Oslo, Norway

[10]IMT Nord Europe, Institut Mines-Télécom, Univ. Lille, Center for Energy and Environment, F-59000 Lille, France

[11]Institute for Ion Physics and Applied Physics, University of Innsbruck, Innsbruck, Austria

[12]Division of Geological and Planetary Sciences, California Institute of Technology, Pasadena, CA, USA

[#] Now at: Institute of Energy and Climate Research, IEK-8: Troposphere, Forschungszentrum Jülich GmbH, Jülich, Germany

[*]Now at: Department of Environmental Science, Policy, and Management, University of California, Berkeley, CA 94720, USA

[**]Now at: 7 and 5

*Correspondence*: Therese S. Carter (tscarter@mit.edu) and Colette L. Heald (heald@mit.edu)

**Abstract.**



Fires emit a substantial amount of non-methane organic gases (NMOGs); the atmospheric
oxidation of which can contribute to ozone and secondary particulate matter formation.
However, the abundance and reactivity of these fire NMOGs are uncertain and historically not
well constrained. In this work, we expand the representation of fire NMOGs in a global chemical
transport model, GEOS-Chem. We update emission factors to Andreae (2019) and the chemical
mechanism to include recent aromatic and ethene/ethyne model improvements (Bates et al.,
2021; Kwon et al., 2021). We expand the representation of NMOGs by adding lumped furans to
the model (including their fire emission and oxidation chemistry) and by adding fire emissions of
nine species already included in the model, prioritized for their reactivity using data from the
FIREX laboratory studies. Based on quantified emissions factors, we estimate that our improved
representation captures 72% of emitted, identified NMOG carbon mass and 49% of OH
reactivity from savanna and temperate forest fires, a substantial increase from the standard model
(49% of mass, 28% of OH reactivity). We evaluate fire NMOGs in our model with observations
from the Amazon Tall Tower Observatory (ATTO), FIREX-AQ and DC3 in the US, and
ARCTAS in boreal Canada. We show that NMOGs, including furan, are well simulated in the
eastern US with some underestimates in the western US and that adding fire emissions improves
our ability to simulate ethene in boreal Canada. We estimate that fires provide 15% of annual
mean simulated surface OH reactivity globally, and exceeding 75% over fire source regions.
Over continental regions about half of this simulated fire reactivity comes from NMOG species.
We find that furans and ethene are important globally for reactivity, while phenol is more
important at a local level in the boreal regions. This is the first global estimate of the impact of
fire on atmospheric reactivity.

# 1 Introduction

Biomass burning (both wildfires and prescribed and agricultural burns) is a large source of non-
methane organic gases (NMOGs) (e.g., Akagi et al., 2011; Koss et al., 2018; Coggon et al., 2019;
Kumar et al., 2018). Goldstein and Galbally (2007) suggest that, while tens of thousands of
organic compounds have been detected in the atmosphere, this may represent only a small subset
of the species present in the atmosphere. Only ~100 compounds have typically been measured
during field campaigns, but recent advances in mass spectrometry have enabled the online





characterization of an expanding suite of organic compounds in the atmosphere, including those
from fires (e.g., Koss et al., 2018). Because many NMOGs are quite reactive, they impact
tropospheric and stratospheric (Bernath et al., 2022) chemistry and composition.  Many NMOGs
are toxic themselves (Naeher et al., 2007), and they can also react to form two major air
pollutants that are also harmful to human health, ozone ($O_3$) and particulate matter under 2.5
microns ($PM_{2.5}$) (e.g., Hobbs et al., 2003; Yokelson et al., 2009; Jaffe et al., 2008, 2013, 2018; Xu
et al., 2021). NMOGs also modulate oxidant concentrations, which affect the climate through the
methane lifetime (Voulgarakis et al., 2013). The importance of fires to the budget of global
NMOGs and to the impacts discussed above is not well understood, as suggested by a recent
study (Bourgeois et al., 2021).

Various terms have been used in the literature to describe reactive carbon-containing trace gases,
including one of the first, non-methane hydrocarbons (NMHCs), which excludes species with
oxygen or other heteroatoms. The term volatile organic compounds (VOCs) encompasses this
broader set of compounds; although, there is no agreed upon, quantitative definition for VOCs or
their surrogate, non-methane organic compounds (NMOCs). The European Union defines VOC
as  any organic compound having an initial boiling point less than or equal to 250° C measured
at a standard atmospheric pressure of 101.3 kPa (European Union, 1999). The US EPA defines
VOCs as any compound that participates in atmospheric photochemical reactions except for
those that they designate as having minimal reactivity. The term oxygenated VOCs (OVOCs)
(Goldstein and Galbally, 2007; Kwan et al., 2006) has further blurred these definitions, with
colloquial usage sometimes being ambiguous as to whether OVOCs are a subset of VOCs or
whether VOCs represent the unoxygenated (i.e., NMHC) suite of compounds. Volatility-based
nomenclature separates VOCs from semi-volatile (SVOC) and intermediate-volatility (IVOC)
species (Robinson et al., 2007). For this study, we use NMOGs, which encompasses all gas-
phase organic compounds (excluding methane), regardless of volatility, degree of oxygenation,
or other chemical properties.

While fires emit a significant amount of NMOGs (> 400 Tg yr$^{-1}$)(Akagi et al., 2011; Yokelson et
al., 2008), second only to biogenic sources globally (~1000 Tg yr$^{-1}$)(Guenther et al., 2012),



modeling efforts, particularly at the global scale, have historically represented only a modest

subset of these emissions and their reactivity. This is in part because a large number of reactive

fire NMOGs remain unidentified (Kumar et al., 2018; Hayden et al., 2022; Akagi et al., 2011).

While progress has been made on measuring emissions of many fire NMOGs, these

measurements have not yet been incorporated into models with global coverage. Given the

significant, but insufficiently characterized variability in emission with both fuel and fire

characteristics, this challenges integration into fire emission inventories. To represent emitted

species, fire emissions inventories generally apply emission factors (EFs) to estimates of dry

matter (DM) burned. Variation among fire inventories is generally driven by differences in DM,

rather than EFs (Carter et al., 2020); though, NMOG EFs often have greater variability amongst

inventories. Akagi et al. (2011) estimated both species-specific NMOC EFs, as well as the EF for

the total of identified + unidentified NMOC mass (for various ecosystems (e.g., for savannas, the

fraction of NMOC emitted mass that is unidentified is ~50% - this number is typical across the

other ecosystems). They also identify unknown NMOCs as one of the largest sources of BB

emissions uncertainties. The GFED version 4 with small fires (GFED4s) inventory (van der Werf

et al., 2017) includes the Akagi et al. (2011) NMOG EFs. The Fire Inventory from NCAR

(FINN) v1.5 also uses the Akagi et al. (2011) species-specific EFs as well as total NMOC, and

total non-methane hydrocarbon (NMHC) EFs (Wiedinmyer et al., 2011). Both the Quick Fire

Emissions Dataset (QFED)(Darmenov and daSilva, 2014) and Global Fire Assimilation System

(GFAS)(Kaiser et al., 2012) rely mostly on an older EF compilation (Andreae and Merlet, 2001)

with a few small updates.


Several recent scientific advances, including a new fire EF compilation, improved

instrumentation, and fire-focused field campaigns, provide opportunities to enhance our

understanding of NMOGs from fires. Andreae (2019) updated the EFs compiled by Akagi et al.

(2011) and Andreae and Merlet (2001) and added 28 more chemical species, including many fire

NMOGs. Recent improvements in instrumentation, especially proton-transfer-reaction time-of-

flight mass spectrometry (PTR-ToF-MS) and gas chromatography (GC), enable high resolution

NMOG measurements, providing the exact molecular formulas and isomer distributions of

detected NMOGs (Hatch et al., 2015; Gilman et al., 2015) and quantification of a substantial

portion of the total carbon mass (Koss et al., 2018). Because OH is generally the dominant



oxidant of most fire NMOGs, the inverse of the OH lifetime (or the OH reactivity, OHR) can be

a useful metric to understand the reactivity of fires, where a gap between summed observed OHR

and calculated OHR based on OH lifetimes can point to unidentified NMOGs or oxidation

products (Yang et al., 2016). Lab studies have shown that, from fires, furans, oxygenated

aromatics, and aliphatic hydrocarbons (e.g., monoterpenes) contribute substantially to both

calculated and measured OHR and that furans and phenolic compounds are among the most

reactive (Coggon et al., 2019; Hatch et al., 2015). The contribution of fires to global OHR has

not been quantified. Growing interest in the impacts of fires on tropospheric composition has

motivated recent fire campaigns in regions with large and growing fire emissions.

These advances suggest that there are opportunities to improve the modeling of NMOGs from

fires and their impacts. In this work, we use the GEOS-Chem chemical transport model (CTM)

and recent lab and field observations to investigate and improve our simulation of fire NMOGs.

We then use this model to characterize the importance of fires to atmospheric reactivity (through

their contribution to total NMOG concentrations and OHR) both globally and regionally.


## 2 Model description

*The GEOS-Chem model*

We use GEOS-Chem (https://geos-chem.org, last access: 15 January 2021), a global CTM, to

explore fire NMOGs globally and in specific large fire regions and outflow regions, such as the

US, boreal Canada, the Amazon, and Africa. GEOS-Chem is driven by assimilated meteorology

from the Modern-Era Retrospective analysis for Research and Applications, Version 2

(MERRA-2), from the NASA Global Modeling and Assimilation Office (GMAO). We use

version 13.0.0 (https://zenodo.org/record/4618180) of GEOS-Chem with a horizontal resolution

of 2° × 2.5° and 47 vertical levels with a chemical time step of 20 min and a transport time step

of 10 min as recommended by Philip et al. (2016). We perform 12-month spin-up simulations

prior to the time periods of interest, June-July 2008, May-June 2012, April-August 2016,

January-December 2017, October 2018, and January-December 2019. We also perform nested

simulations over North America at 0.5° × 0.625° (with boundary conditions from the global



simulation) for comparison against DC3, FIREX-AQ, and ARCTAS observations (see Section 4)
with chemistry and transport time steps of 10 and 5 min, respectively.

GEOS-Chem includes $SO_4^{2-}/NO_3^-/NH_4^+$ thermodynamics (Fountoukis and Nenes, 2007)
coupled to an $O_3$–VOC–$NO_x$–oxidant chemical mechanism (Chan Miller et al., 2017; Mao et al.,
2013; Travis et al., 2016) with integrated Cl-Br-I chemistry (Sherwen et al., 2016). We add
aromatic oxidation updates (with benzene, toluene, and xylenes (C8 aromatic compounds
including o-, m-, p- xylenes and ethylbenzene) emissions) per Bates et al. (2021) and ethene and
ethyne chemistry updates per Kwon et al., (2021); both were developed in GEOS-Chem, but not
yet implemented in the standard model. These aromatic and ethene/ethyne chemistry updates
modify oxidant levels, particularly $NO_3$, which overall decreases NMOG lifetimes. Bates et al.
(2021) estimate an annual global mean increase of +22% for $NO_3$. In general, species not directly
involved in the new chemistry are modestly impacted by these changes while, for example,
species like glyoxal and glycolaldehyde, which are important products of the ethene/ethyne
chemistry, undergo large increases.

Baseline fire emissions are from the Global Fire Emissions Database version 4 with small fires
(GFED4s; (van der Werf et al. 2017)) and are specified on a daily timescale. Additional details
on fire NMOG emissions are provided in Sect. 3. A sensitivity analysis, described in Sect. 4,
uses FINNv1.5. Anthropogenic emissions (including fossil and biofuel sources) follow the year-
specific CEDS global inventory (Hoesly et al., 2018). Trash burning emissions are from
Wiedinmyer et al. (2014). Aircraft emissions are from the Aviation Emissions Inventory Code
(AEIC) inventory (Stettler et al. 2011; Simone et al. 2013). Biogenic emissions are calculated
online from the MEGANv2.1 emissions framework (Guenther et al. 2012).

A typical source attribution method in models zeroes out a specific source and differences that
simulation from the baseline. This brute force method is ideal for linear systems, but for non-
linear chemistry, large perturbations to emissions will feed back onto the chemistry (and thus
impact lifetimes). For example, zeroing out fire emissions increases OH concentrations because





the OH sink has been decreased, thereby increasing the rate of oxidation of other species, such as from biogenic sources. Such a depression in isoprene concentrations, for example, may then

increase or decrease ozone concentrations, depending on the chemical regime. The HTAP modeling experiments, which were focused on $O_3$, address this issue with 20% emission perturbation sensitivity studies – a number chosen to produce a discernable (larger than numerical noise) and realistic impact while minimizing non-linearities (Huang et al., 2017). To isolate the influence of fires in our model and minimize these nonlinearities, we run emissions

sensitivity simulations with 5% more and less fire emissions (0.95 and 1.05 times fire emissions) and scale up the difference to equate to a 100% perturbation. We compare these runs with the more typical noFires brute force simulation in the SI (see Figs. S1, S2, and S3) and show, for example, that the $O_3$, OH, and isoprene differences are minimized with the emissions sensitivity approach (Fig. S1). We use this fire sensitivity source-attribution approach throughout this study.


To translate the concentrations of reactive compounds to calculated OHR (cOHR) at atmospheric ambient conditions, we define cOHR as the sum of the pressure- and temperature-dependent OH rate constant of a species (from the GEOS-Chem mechanism) with its concentration as follows:

$$cOHR\ (s^{-1}) = k_{OH,CH_4}[CH_4] + k_{OH,CO}[CO] + k_{OH,NO_2}[NO_2] + \Sigma k_{OH,NMOG_i}[NMOG_i] + \cdots \ (1)$$

where $i$ indicates various NMOG species.

## 3 Updating and expanding fire NMOGs in GEOS-Chem

We update and expand the fire NMOGs in GEOS-Chem by updating existing EFs and then considering additional emissions and chemistry. First, we update our EFs from Akagi et al.

(2011) to the newer Andreae (2019) compilation. Total NMOG emissions do not change substantially between the two inventories – in 2019, they decrease by 3.4% from Akagi et al. to Andreae. There is, however, more variation across the different species with, for example, Akagi et al. providing larger savanna EFs for glycoaldehyde (0.25 g/kg DM vs. 0.13 g/kg DM) and glyoxal while Andreae specifies higher values for benzene (0.33 g/kg DM vs. 0.20 g/kg DM),

toluene, and xylenes (see Fig. S4 in the SI).



The standard GEOS-Chem model includes fire emissions of 15 NMOG species. The number of possible additional NMOGs from fires is quite large (Akagi et al., 2011). We focus on the feasibility and utility of adding fire NMOGs that Coggon et al. (2019) (building on Koss et al.,

(2018)) identify as accounting for 95% of fire OHR. We first identify the fire NMOGs already represented in GEOS-Chem (black circles in Fig. 1), then those additional species for which EFs are available from the recently updated compilation by Andreae (2019) in blue, and finally those species for which EFs are only available for western US fuel types as measured during the FIREX lab study (Koss et al., 2018) in red. We size the symbols in Fig. 1a by their EFs for

savanna (EFs for other fuel types generally provide a similar relative ranking) to identify the largest NMOG emissions. We order the species in Fig. 1a by their decreasing lifetime against OH with values ranging from 1 hour for sesquiterpenes to over a month for ethane. For context, we provide the same plot by their lifetimes against two other important oxidants, $O_3$ and $NO_3$, in the SI (Fig. S5). To explore how chemical lifetimes of these fire NMOGs compare with their

physical lifetime in a model grid box, we estimate the approximate lifetime of transport out of a global $2° × 2.5°$ grid box (~20 hours) and for a nested grid box at $0.5° × 0.625°$ (~5 hours) using 3 m/s as the surface wind speed (shown as the grey shaded region).

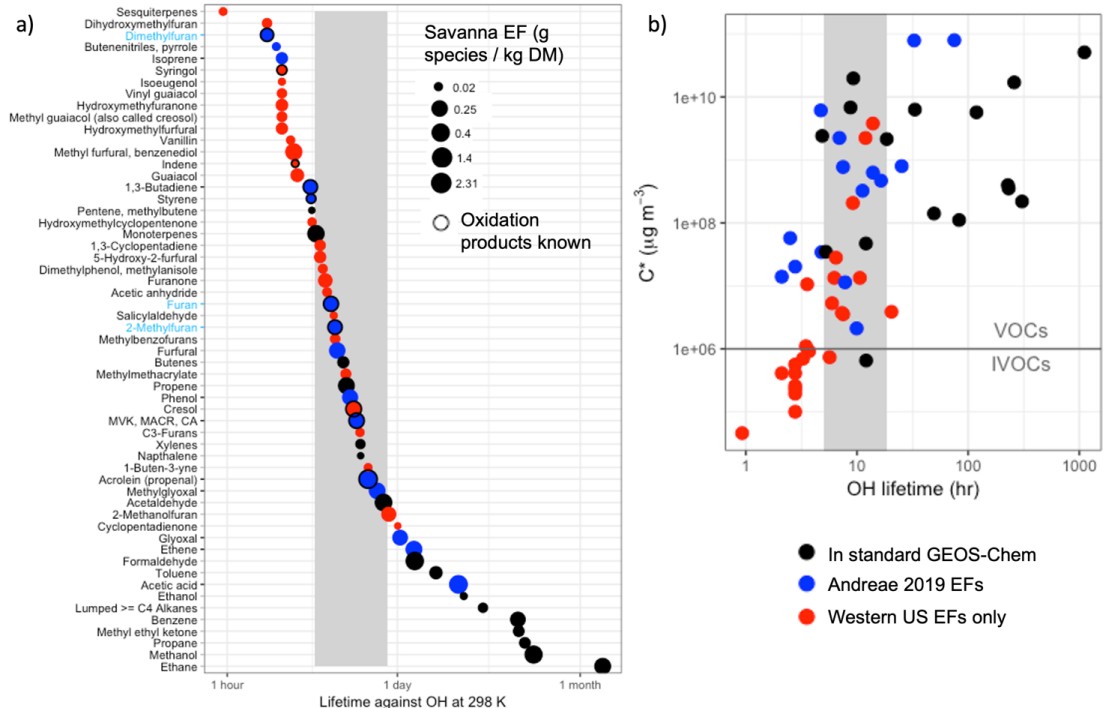

*Figure 1. (a) NMOGs emitted from fires, shown in descending order of chemical lifetime due to oxidation by OH (at 298 K). Only the species responsible for 95% of OHR from fires are shown (following Coggon et al., 2019). NMOGs included in the standard GEOS-Chem model are in black, species not included in the standard GEOS-Chem model but where emissions factors are available in Andreae (2019) are in blue, and species that are only available for western US fuel types from Koss et al. (2018) are in red. The y-axis tick marks are in black for fire NMOGs added to GEOS-Chem in this study and in blue (with blue labels) when both the fire NMOG and its oxidation chemistry were added. The points are sized by their relative savanna and grassland (labeled "savanna") emission factor in g species / kg DM burned. The grey vertical box represents an approximate physical lifetime against transport out of a nested 0.5° × 0.625° grid box (~5 hours) and a 2° × 2.5° grid box (~20 hours). CA stands for crotonaldehyde. (b) Plot of volatility (C\*) against OH lifetime for the species shown in (a) using the same color conventions. The horizontal line separates VOCs from IVOCs based on their C\*.*

In Fig. 1 most species with chemical lifetimes that exceed the transport timescale out of a model grid box are already included in the model. Using Andreae (2019) EFs, we add fire emissions of eight species already included in the model for which fire emissions were previously neglected: phenol, methyl vinyl ketone (MVK), ethene, isoprene, acetic acid, methylglyoxal, glyoxal, and lumped aldehydes with three or more carbon atoms, which does not include furfural (RCHO). We also add fire emissions of 1,3-butadiene to the tracer representing alkenes with greater or equal to three carbons (PRPE). Furans from fires are important for atmospheric reactivity (Koss et al., 2018; Coggon et al., 2019). We add a new lumped furan tracer, called FURA, that combines the pyrogenic emissions of furan, 2-methylfuran, and 2,5-dimethylfuran and uses the





OH rate constant of furan ($k_{OH}$ = 1.32 x $10^{-11}$ x $e^{\frac{-334}{RT}}$) (furan and 2-methylfuran dominate

emissions and have very similar lifetimes against OH). In the model, the oxidation of FURA with OH produces butenedial since that has been shown experimentally with an estimated carbon balance of 100% C (Bierbach et al., 1995). Thus, we add fire emissions for almost all the species for which we have Andreae EFs (12 species) to GEOS-Chem. For 2019, these added global fire emissions (19.6 Tg C) are roughly equivalent to the fire NMOG emissions already in the model

(21.8 Tg C) (see Table S1 for species total emissions). The only species with Andreae (2019) EFs that we do not add to GEOS-Chem are: (1) butenenitriles, which have a very small EF and a short lifetime against OH, (2) styrene, which also has a chemical lifetime less than the grid box physical transport time, and (3) furfural. There is a wide spectrum of lesser abundant furans (+ furfural) (Zhao and Wang, 2017) that contribute to furan reactivity; therefore, the representation

in this model constitutes a lower bound on furan contributions to total reactivity. We do not include species where EFs are only available for western US fuel types from Koss et al. (2018). Figure 1a suggests that nearly all these species are very reactive and short-lived as evidenced by the red circles being within or below the physical transport time of the grid box.

Fig. 1b shows the volatility of these same NMOG species. The species for which we have global EFs available are almost entirely very volatile and above a commonly held cutoff threshold for intermediate volatility compounds (IVOCs) versus VOCs (C*=1 × $10^6$ μg m$^{-3}$ (Ahern et al., 2019 and references therein)). This suggests that both the standard model and our expanded treatment of NMOGs neglect many NMOG precursors for secondary organic aerosol. This study focuses

on the OHR of NMOG from fires; further work is needed to constrain the EFs (Fig 1b. suggests that global EFs are not available for most IVOCs) and oxidation chemistry of NMOG species relevant to SOA formation from fires.

Species whose chemical lifetimes are shorter than the physical transport lifetimes (Fig. 1a) contribute strongly to near-field reactivity but are likely not exported from the grid box of emission. For these species, oxidation rapidly converts emitted species into secondary products, and it is these products that are exported away from the fire source. However, a detailed knowledge of the oxidative chemistry of many of these species is lacking (as evidenced by the





small number of black circled species, indicating "oxidation products known" in Fig. 1a). In
particular, we note that we do not include several very reactive species (e.g., furfurals, guaiacol)
(Coggon et al., 2019). Hence our model represents a lower limit of reactivity from fires, despite
our inclusion of longer-lived NMOG.

To characterize the amount of carbon mass and reactivity represented in our current model and
the potential shortfall in NMOG emissions, we use EFs as proxies for emissions. We first
calculate the total carbon mass based on the sum of the savanna and temperate forest EFs (the
only EFs we have for all species in Fig. 1), and we compare that number to the sum of the
savanna and temperate forest EFs for different subsets (standard GEOS-Chem and updated
GEOS-Chem) of species included in Fig. 1. We note that here and throughout the manuscript,
NMOG % values refer to percentage by carbon mass.We find that the standard GEOS-Chem
model represents 49% of the total carbon mass emissions potential of NMOGs suggested in Fig.
1. Our additions to the model increase this to 72%. We then multiply these EFs by the rate
constants with OH at 298 K to represent a proxy for reactivity. From this, we calculate that the
standard model includes 28% of the potential emitted reactivity of savanna and temperate forest
fuel type emissions; our model updates add an additional 17% (for a total of 45% of the potential
reactivity). This suggests that the sum of these minor species for which global EFs are not
defined, and therefore that we do not include in our model, contribute over half of the emitted
reactivity from fires. We note that all of these fractions are relative to speciated NMOGs from
Coggon et al. (2019); unspeciated or unidentified NMOG would increase our model shortfall.


We use the Andreae (2019) EFs applied to the GFED4s DM and the chemistry updates noted
here for the rest of this analysis unless specifically noted.

## 4 Exploring observational constraints on fire NMOGs

There are limited observational constraints on fire-influenced NMOGs and OHR. We use
observations of OHR made at the Amazon Tall Tower Observatory (ATTO) and of VOCs from





the FIREX and ARCTAS campaigns in addition to measurements of both VOCs and OHR from the Deep Convective Clouds and Chemistry (DC3) campaign. Previous work has shown that the plume-chasing sampling strategy of the WE-CAN 2018 field campaign limits the suitability of this dataset for 3D model evaluation (Carter et al., 2021). While the KORUS-AQ campaign included airborne OHR measurements and some fire influence (median concentration of acetonitrile, a biomass burning tracer (Lobert et al., 1990), ~165 ppt, Fig. S6), the campaign is dominated by anthropogenic sources, which recent work shows may confound the acetonitrile signal (Huangfu et al., 2021); and therefore we do not include this campaign in our analysis. We explore observations of OHR taken during ATom-1 off the coast of western Africa, during which Strode et al. (2018) identified fire influence. However, the aircraft sampled air masses more than 3000 km away from the continental fire source. As a result, most short-lived NMOG have reacted away, and the modeled cOHR is low and dominated by CO (Fig. S7). Thus, ATom-1 is not a good constraint on fire cOHR and the impact of NMOG. There are no other airborne campaigns that we are aware of that have deployed OHR instrumentation in fire-influenced environments.

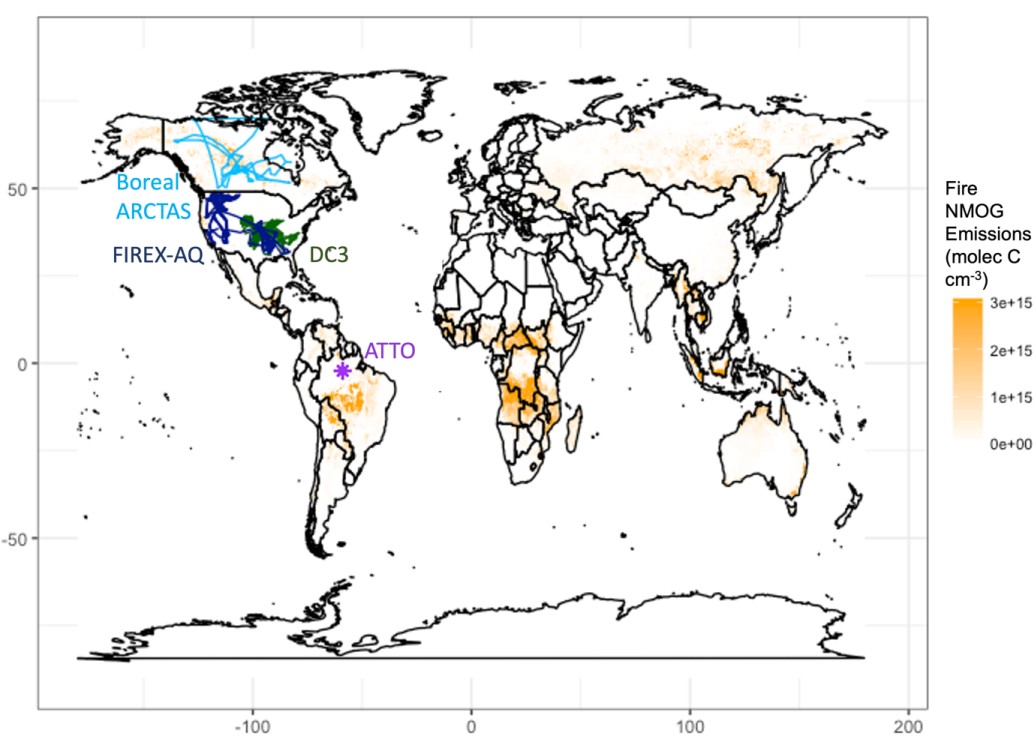



*Figure 2. Measurement locations of campaigns used in this analysis overlaid on annual mean NMOG emissions from fires across 1997 – 2019 from GFED4s. Boreal ARCTAS is in light blue, FIREX-AQ in dark blue, DC3 in dark green, and ATTO in purple.*

We use observations from campaigns that sampled fire-influenced air masses in different regions around the world (Fig. 2). For tower and aircraft campaigns, the model is sampled to the nearest grid box of the measurements both temporally and spatially using the entire 1-min merge of observational data. We then average both the model and the observations to the model grid box.

To evaluate our simulation of NMOGs in the US, we explore observations from the NASA DC-8 during the Fire Influence on Regional to Global Environments and Air Quality (FIREX-AQ) campaign, which deployed in the western and eastern US from 15 July through 5 September 2019 with a large suite of NMOG instrumentation aboard. The campaign investigated the chemistry and transport of smoke from both wildfires and prescribed burns in the western and

eastern US with flights originating from both Boise, ID, and Salina, KS. CO was measured using a modified commercial off-axis ICOS instrument (Los Gatos Research (LGR) $N_2O$/CO-30-EP; Baer et al. 2002) at ~ 4.6 μm. Precision was estimated to be 0.4 ppb, and uncertainty for the dry air mole fraction of CO for mixing ratios below 1 ppm to be ± (2.0 ppb + 2%). More details are available from Bourgeois et al. (2022). MVK, furan, 2-methylfuran, and 2,5-dimethylfuran were

measured using the NCAR Trace Organic Gas Analyzer with a Time-of-Flight MS (TOGA-TOF) (Apel et al., 2015; Wang et al., 2021). The TOGA-TOF measurements are reported with a detection limit of 0.5 ppt and an uncertainty (accuracy and precision) of 20%. Phenol was measured using the NOAA PTR Time-of-Flight MS (PTR-ToF-MS) with accuracy of 25% (Müller et al., 2014; de Gouw and Warneke, 2007) and by the California Institute of Technology

Chemical Ionization Mass Spectrometer (CIT-CIMS) with an accuracy of 30% (Xu et al., 2021). During the western part of the campaign, the phenol measurements by PTR were affected by a contamination issue above 8 km, so those data have been removed. Generally, the model captures the differing fire influence in the eastern and the western US. For example, in the eastern US, the model captures vertical profiles of CO well (Fig. 3) while in the western US, the

model matches the general shape but underestimates the magnitude of the observations and likely the influence of more sporadic fires in the region. Recent papers have also shown that GEOS-Chem struggles to fully capture large wildfires in the western US (e.g., Carter et al., 2021; O'Dell et al., 2019; Zhang et al., 2014) in part because the DM estimates may be underestimated



(Carter et al., 2020) and because GEOS-Chem and other air quality models with a fairly coarse
resolution have trouble resolving sub grid processes (Eastham and Jacob, 2017; Rastigejev et al.,
2010), including those involved in fire plumes (Wang et al., 2021; Stockwell et al., 2022).

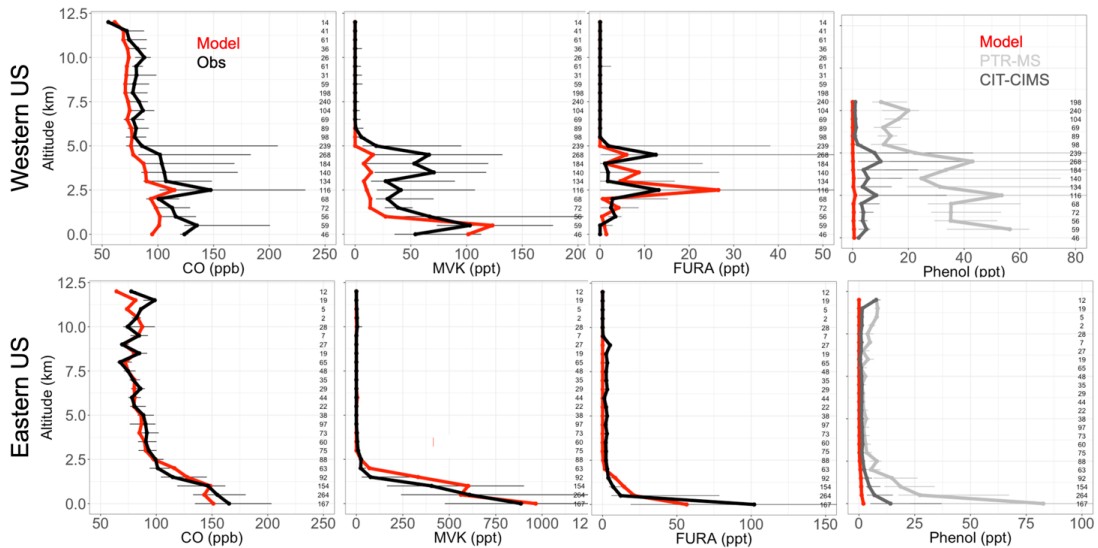

*Figure 3. Binned observed (black) and simulated (red) median vertical profiles of CO, MVK, lumped furan species (FURA =
furan + 2-methylfuran + 2,5-dimethylfuran), and phenol concentrations from the FIREX-AQ campaign. For phenol, observations
using the PTR-MS are in light grey and those from the CIT-CIMS in dark grey. Horizontal bars show the 25th–75th percentile
range of measurements in each vertical 0.5-km bin. The number of observations in each bin is shown on the right side of each
panel.*

The FIREX-AQ measurements can also be used to evaluate some of our model updates. Figure 3
shows that MVK, an example NMOG for which we added fire emissions in the model, follows

similar model performance as CO. We note that more than 80% of simulated MVK during
FIREX-AQ comes from fires. The FIREX-AQ summed observations of the same three furan
species suggest that our new lumped "furan" (FURA) tracer with only fire sources performs well
in the eastern and western US (Fig. 4). This suggests that the furan EFs for US fires like those
sampled are accurately captured in the Andreae (2019) compilation; although they may also be

overestimated and thus compensating for an underestimate in the DM burned in GFED4s. Fig. 3
shows that our addition of fire emissions of phenol still underestimates observed concentrations
across all altitudes in the western US and at the surface in the eastern US. Phenol observations
from the CIT-CIMS (dark grey) instrument are a factor of 3 lower than the PTR-MS (light grey).
The model underestimates the lower phenol concentration (CIT-CIMS) by a factor of 8 in the

eastern US and 15 in the western US. Given that both instruments were calibrated for phenol, the



diference between two measurements is not yet accounted for. The measurements of phenol and other less-studied compounds have substantial uncertainties as indicated by these instrument differences, and more work is needed to understand these uncertainties. However, Taraborrelli et al. (2021) suggest that anthropogenic and fire sources contribute roughly equally to phenol

emissions at the global scale. Therefore, both higher phenol emission factors from fires and emissions  from anthropogenic sources in the US are likely needed to help resolve the discrepancy seen in Fig 3.

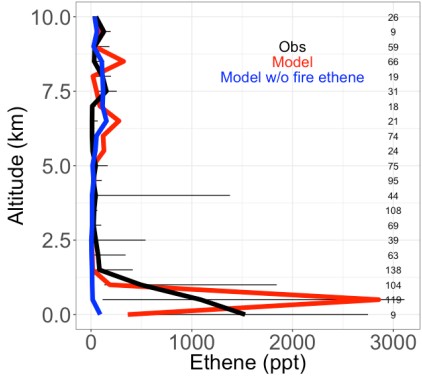

*Fig 4. Median vertical profiles of binned ethene mixing ratios, including observed (black), simulated (red), and simulated in a*
*sensitivity model run without fire ethene emissions (blue) during the boreal part of the ARCTAS campaign. Horizontal bars show the 25th–75th percentile range of measurements in each vertical 0.5-km bin. The number of observations in each vertical bin is shown on the right side of each panel.*

In this study, we add fire emissions of ethene to the model, which may be important in certain regions. We turn to the boreal component of the Arctic Research of the Composition of the

Troposphere from Aircraft and Satellites (ARCTAS) campaign to test the fidelity of this addition because the boreal EFs for ethene are high (1.54 g/kg DM, compared to 0.83 g/kg DM for savanna) and there is less anthropogenic influence in the boreal regions to confound a fire-focused model evaluation. The ARCTAS campaign sampled the Arctic region with an emphasis on forest fire smoke plumes using the NASA DC-8 aircraft from 18 June to 13 July 2008 (Jacob

et al., 2010). We use observations of ethene from the UC Irvine Whole Air Sampler (WAS). This measurement has a limit of detection of 3 ppt, 3% precision, and 5% accuracy. See Simpson et al. (2011); Colman et al. (2001) for more details. We show that the model (in red), filtered to remove the least fire-influenced points, captures the observed vertical profile of ethene concentrations well, including the large enhancement at the surface (Fig. 4). This is an





improvement over a simulation without fire emissions of ethene (shown in blue), which shows
        negligible ethene throughout the vertical profile.

        Figure 5 compares OHR measurements made at the ATTO site during the fire seasons in October
        2018 and September 2019 (Pfannerstill et al., 2021) with the updated GEOS-Chem model
simulation. ATTO is situated ~150 km northeast of Manaus, Brazil. We use total OHR
        measurements taken at 80, 150, and 320m on the tower during two intensive observation periods
        in October 2018 and September 2019 using the Comparative Reactivity Method (CRM, Sinha et
        al., 2008), which is described in more detail by Pfannerstill et al., (2021). We confirm that
        simulated OHR is mostly driven by isoprene during this campaign as Pfannerstill et al. (2021)
show for the observations and find that the model (in red; median = 21.5 s$^{-1}$) captures the overall
        observed (in black; median = 22.4 s$^{-1}$) cOHR (Fig. 5a). Pfannerstill et al. (2021) assessed that
        fires contribute 17% of their OHR measurements (shown in black in Fig. 5b). Our updated
        simulation with the Andreae (2019) EFs and new chemistry (red in Fig. 5b) underestimates this
        fire contribution by a factor of ~5. Reddington et al. (2016, 2019) suggested that the FINN1/1.5
and GFED3/4s fire inventories underestimate fire emissions by a factor of 2-3 in parts of the
        Amazon with FINN emissions generally less biased than GFED. Following their analysis, we
        perform a sensitivity simulation where we use FINN1.5 instead for fire emissions and scale up
        the emissions to match what was used in the Reddington et al. studies. This simulation (purple)
        greatly improves model-observations agreement with a mean fire cOHR contribution of 17%
(Fig. 5b). Excluding our additions to the NMOG model description (purple and white hatching)
        does not substantially degrade the agreement with observed OHR from fires, suggesting that
        underlying biomass burned may be a more important uncertainty in fire NMOG OHR than
        missing reactive species in this region.

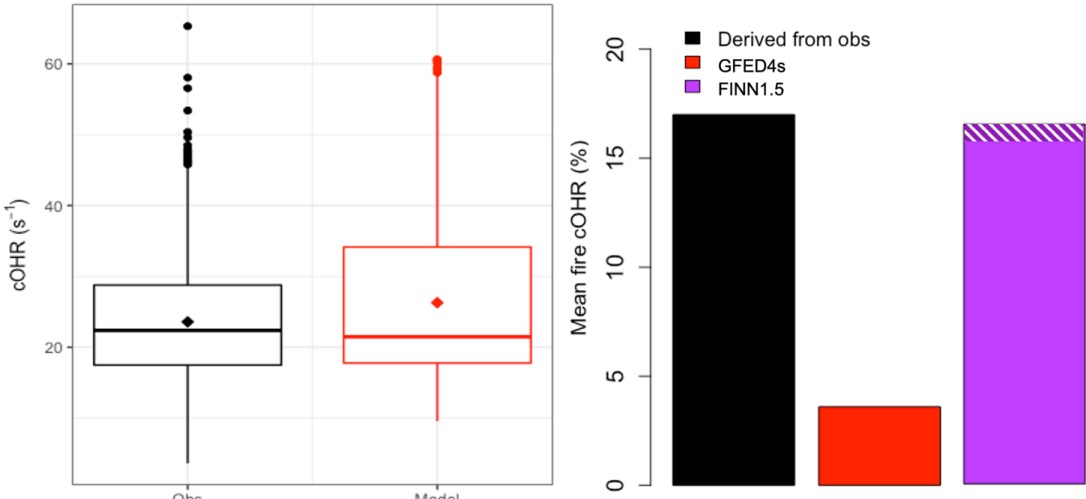

*Figure 5. (Left) Boxplots of cOHR during the October 2018 and September 2019 measurement periods at ATTO with observations in black and the model in red with medians shown as a horizontal line and means as diamonds. (Right) The mean percentage contribution from fires during the same time period to cOHR with that derived from observations in black, the model with GFED4s in red, and a simulation using scaled-up FINN1.5 in purple. The white hatching on the FINN1.5 simulation indicates the increase in percentage contribution due to of new EFs and chemistry.*

We also compare observations of NMOGs and OHR during the DC3 campaign, which sampled in the southeastern and south central US in 18 May – 22 June 2012 (Barth et al., 2015). Acetonitrile was measured using a PTR-MS (Hansel et al., 1995; Wisthaler et al., 2002). The OHR measurement is described in detail in Brune et al., (2018); Mao et al., (2009) with a limit of detection for 20s measurements estimated to be ~ 0.6 s$^{-1}$. This campaign was influenced by numerous sources, including fires. Here we explore how well GEOS-Chem captures observed OHR as a function of fire influence. Figure 6 shows the model skill in reproducing OHR (model minus observations) against CO and acetonitrile. We find that model skill degrades generally monotonically with increasing acetonitrile and CO concentrations. No similar trend is observed with anthropogenic tracers such as benzene, suggesting that the model underestimates fire sources of reactivity. This confirms that we are likely missing emitted fire NMOGs and/or secondary products during this campaign beyond what we is currently represented in the model, as suggested in Section 3. Thus, while previous comparisons shown in Section 4 indicate that the additions we have made to the model have improved our simulation of fire NMOGs, Figure 6 confirms further work is needed to fully capture the impact of fires on OH reactivity.



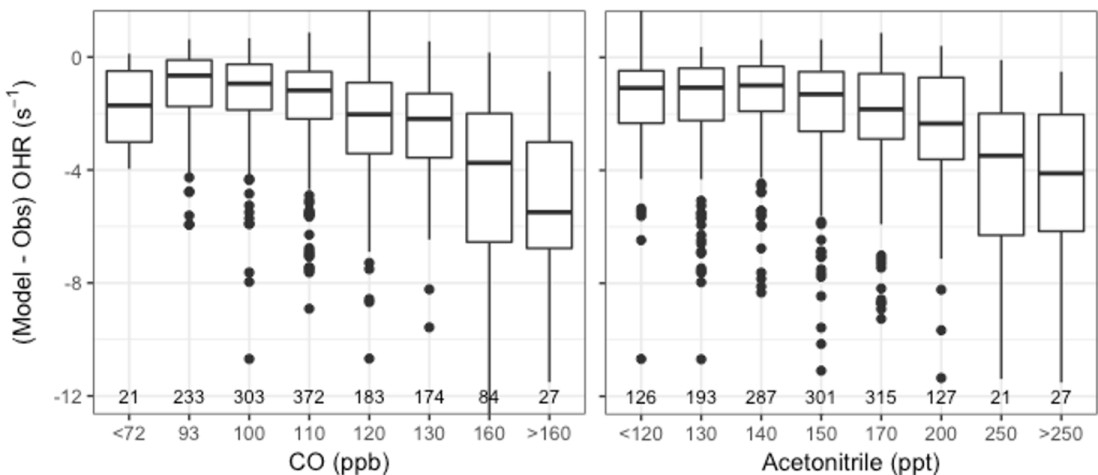

*Figure 6. Boxplots of the model minus observed OHR or OHR difference during the DC3 campaign against binned CO (left) and acetonitrile (right) observations. The number of observations in each bin is shown on the bottom of the panel.*

## 5 Characterizing fire contribution to global NMOG and atmospheric reactivity

The first estimates of global simulated cOHR highlight the strong gradients in reactivity from source regions to background (Safieddine et al., 2017; Lelieveld et al., 2016). To date, there has been no effort to attribute simulated cOHR to sources. Here we use the source-attribution approach described in Sect. 2.1 to assess the contribution of fire emissions to global NMOG and cOHR. We note that, given the discussion of Sect. 3, this global simulation should be taken as a lower limit for fire NMOG and cOHR, particularly in fire source regions.

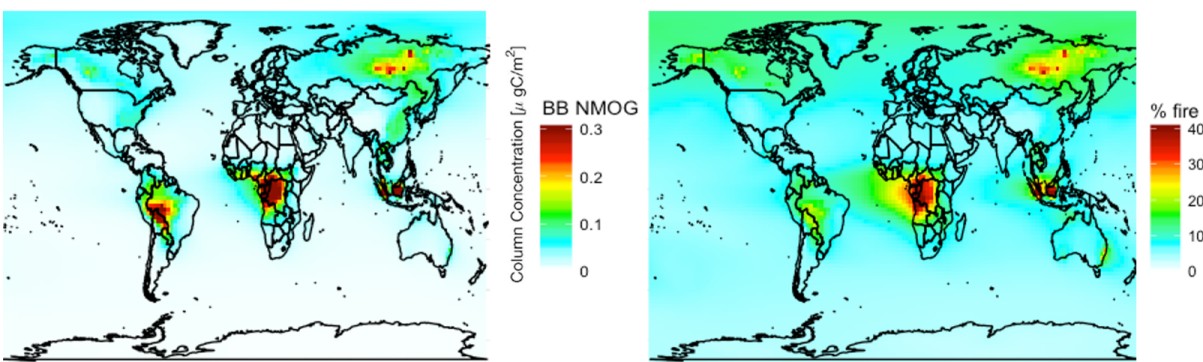





*Figure 7. 2019 annual mean simulated (Left) NMOG total column concentrations from fires and (Right) percent contribution of NMOG total column concentrations from fires*

Figure 7 shows simulated annual mean 2019 NMOG total column concentrations and percent contribution from fires. Fires exceed 40% of NMOG annually, not just in the fire seasons, in several fire source regions (e.g., Siberia, central Africa, and Southeast Asia) with elevated levels (~25%) across large parts of the northern hemisphere downwind of sources. Fires also contribute

more than 5% of NMOGs nearly everywhere globally, including the remote ocean, driven by RCHO, acetaldehyde, ethene, propene, and DMS.

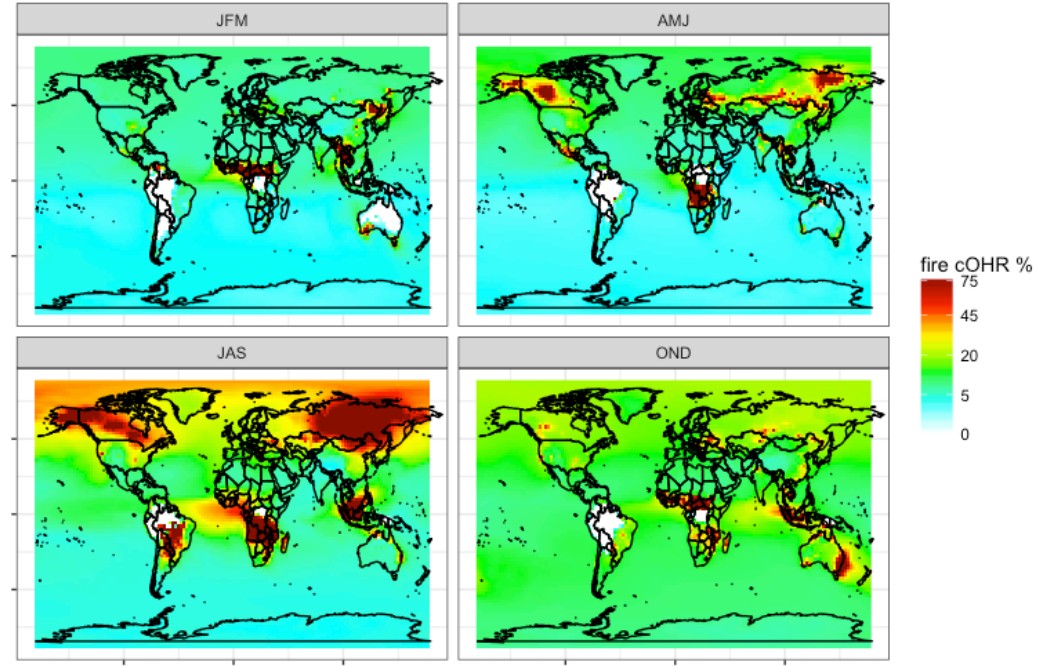

*Figure 8. Percent surface cOHR seasonally from fires in 2019 from updated GEOS-Chem simulation (see Sect. 3) using GFED4s DM with Andreae (2019) emission factors. JFM is January, February, March; AMJ is April, May, June; JAS is July, August, September; and OND is October, November, December.*


Figure 8 shows that the contribution of fires to seasonal surface cOHR in 2019 is substantial, exceeding 75% in large fire source regions. The large fire contribution in July, August, September (JAS) and, to a lesser extent, in other seasons, contributes to cOHR beyond the immediate fire emission region. We note that these values are year dependent, and, for example,

2019 was a low fire year in the western US where we might expect a larger fire contribution in other years (see Fig. 12 for more discussion of interannual variability). Longer-lived fire species (particularly CO) contribute 10-25% of the background cOHR, peaking in October, November,



and December (OND). Globally in 2019, the annual average simulated fraction of surface

reactivity from fires is 15%. The relative export of OH reactivity from a fire source regions is

expected to vary with the mix of emissions (i.e. chemical reactivity) and the oxidative

environment. This can be explored in fire-dominated regions with strong zonal winds, which

produce a clear fire plume. Fig. S8 suggests that the cOHR from fires decays more slowly in

plumes from boreal source regions (Canada and Siberia) compared to the tropics (Central

Africa), likely reflecting differences in the oxidative loss.

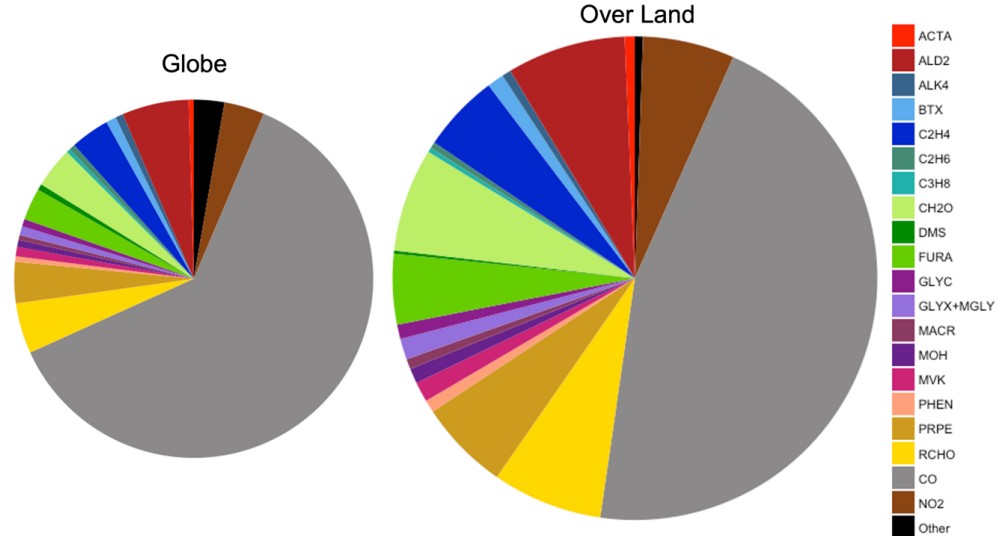


*Figure 9. Average global annual mean contribution of chemical species and species groups to simulated surface fire cOHR in 2019 using GFED4s DM for the entire globe and over land only. The two pie charts are approximately sized by their relative fire cOHR (0.16 s$^{-1}$ globally and 0.29 s$^{-1}$ over land). BTX is benzene, toluene, and xylenes. Other(n=56) includes species where their annual mean fire cOHR is equal or less than 0.0004, which includes ozone.*

Figure 9 shows that NMOGs make up 48% of the annual mean surface fire cOHR over land (and

33% over the whole globe), with CO and $NO_2$ providing the bulk of the remaining cOHR. Of the

non-CO annual mean surface fire cOHR, NMOGs make up roughly 90% (colors in Fig. 9).

Particularly important NMOG contributors to fire reactivity include acetaldehyde (ALD2 in dark

red; 15% of non-CO fire cOHR), formaldehyde ($CH_2O$ in light green; 13% of non-CO fire

cOHR), and fire emissions of several NMOG species added in this work – lumped furan (FURA

in lime green; 9% of non-CO fire cOHR), ethene ($C_2H_4$ in royal blue; 10% of non-CO fire

cOHR), propene and higher carbon alkenes (PRPE in tan; 11% of non-CO fire cOHR), and





lumped aldehydes greater than or equal to three carbons (RCHO in yellow; 14% of non-CO fire cOHR).

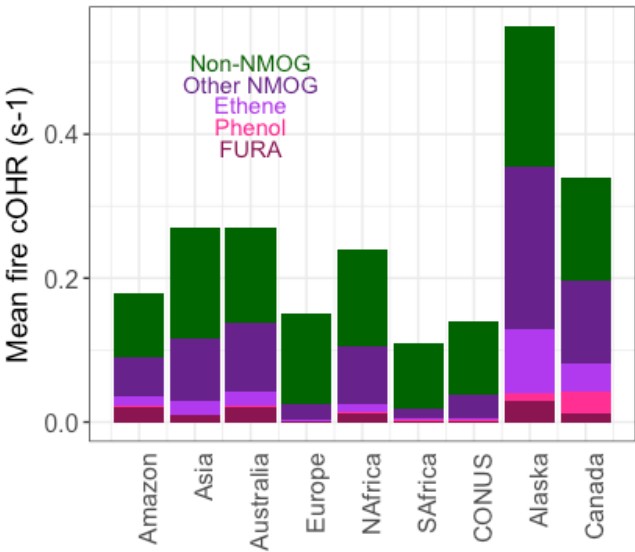

Figure 10: Mean simulated surface fire cOHR in 2019 using GFED4s DM regionally for the Amazon, Asia, Australia, Europe, northern hemispheric Africa (NAfrica), southern hemispheric Africa (SAfrica), the contiguous US (CONUS), Alaska, and Canada.

Because fires and fuel types differ regionally, Fig. 10 shows the simulated annual mean fire cOHR in several large fire regions. The addition of fire emissions of lumped furans, phenol, and ethene contribute significantly to fire cOHR depending on region, consistent with their EFs. Lumped furans, "FURA" (dark red), contribute the most in the Amazon (11%), Australia (7%), and Alaska (5%) and the least in Europe (1%), southern Africa (2%), and CONUS (2%). The boreal regions (Alaska and Canada) show larger contributions from phenol (bright pink) (2% and 9%, respectively) and ethene (light purple) (16% and 12%, respectively) consistent with high boreal EFs. Other NMOGs (dark purple) also contribute substantial cOHR in most regions except Europe, southern Africa, and CONUS where the contribution from CO is dominant. We note that given our observational analysis for the Amazon (Fig. 5), fire emissions, and thus the fire cOHR, in this region, and possibly other tropical regions, are likely drastically underestimated in our simulation. We do not adjust regional fire emissions here given that large uncertainties remain on fire emissions in the Amazon and tropics more generally; therefore, the





values shown in Fig. 10 should certainly be considered a lower estimate (in the Amazon by more
than a factor of 3, following Fig. 5).

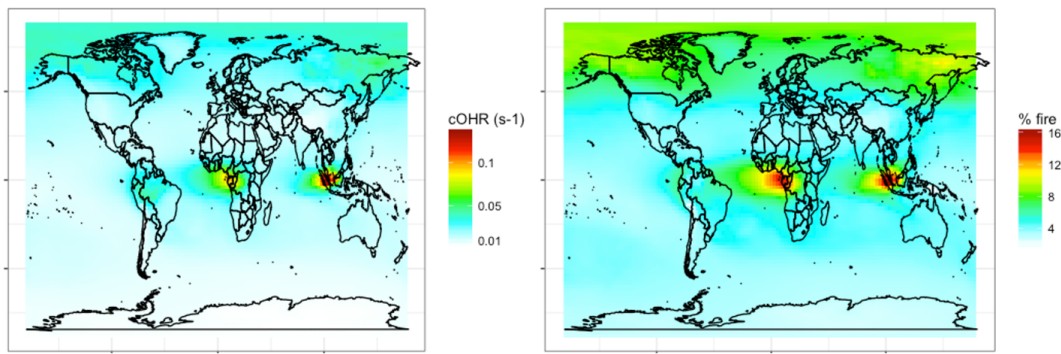


*Figure 11: Annual mean simulated (left) fire cOHR and (right) percent of cOHR from fires using GFED4s DM in 2019 at 500 hPa.*

NMOGs and associated reactivity in the free troposphere are relevant to the global oxidative
capacity, long-range transport, and climate. Figure 11 shows that the simulated contribution of

fires in the mid troposphere (500 hPa) to cOHR is 5% globally, but reaches ~15% in the tropics.
Fires also contribute more reactivity (~10% annually) in the boreal region. We undertake a
similar analysis for 2017 (not shown) where the magnitudes and spatial trends discussed in Figs.
7-11 are similar.

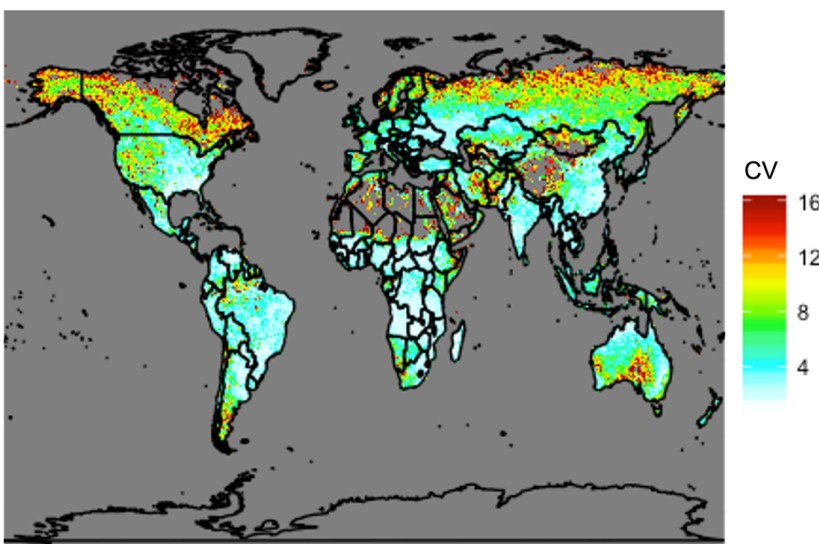



*Figure 12. Coefficient of variation (CV) of total annual carbon emissions from fires from 1997 to 2019 using GFED4s DM. Coefficient of variation is defined as the standard deviation of a quantity divided by the mean, which is a statistical measure of the relative dispersion of the dataset about the mean.*

The analysis presented above is for a single year (2019). Fire location and magnitudes vary
substantially year to year. The global total carbon emissions of NMOGs from the GFED4s inventory from 1997 to 2019 range from 27 to 48 Tg C yr$^{-1}$ with 41 Tg C emitted in 2019. This suggests that our estimates of fire's contribution in the preceding analysis is representative of average conditions at the global scale but may increase or decrease by roughly a third in different years. Therefore, across years, the annual global average fraction of surface reactivity due to
fires likely ranges from ~10 – 20% with large uncertainties due to the magnitudes of other anthropogenic and biogenic emissions in any given fire region. To understand interannual variability at a more local scale, Fig. 12 shows the coefficient of variation, a statistical measure of the relative dispersion of the data about the mean, for total carbon emissions from fires across the same years. Given their propensity for large wildfires, the boreal regions, the western US,
and Australia show greater year to year variability, which would translate to high variability in fire contributions to surface cOHR. Conversely, Africa shows very little variation consistent with human-ignited savanna and agricultural burning each year, suggesting that our single year estimates of fire contributions to cOHR are generally robust in this region.

## 6 Conclusions

Recent work has suggested that NMOGs from fires may be a large source but noted that we did not yet have a framework in our models to fully characterize them and their reactivity (Akagi et al., 2011). Our work provides a first estimate of fire NMOGs globally and regionally and their contribution to reactivity. We updated fire NMOG EFs to Andreae (2019) from Akagi et al.
(2011). We also expanded the model representation by adding new fire NMOGs (e.g., lumped furans, phenol, ethene), prioritized for their reactivity using data from the FIREX lab studies and their chemistry. We used a suite of recent observations from the lab (FIREX) to towers (ATTO) to aircraft campaigns (FIREX-AQ, ARCTAS, DC3) to constrain and test our model representation. We show that observations support the additions made to the model.




Our model suggests that fires are a major contributor to NMOG concentrations, especially near fire source regions and downwind of them. We show that fires provide more than 75% of cOHR in large parts of the northern hemisphere and that fires contribute to a high background (~25%) reactivity beyond their source regions, mostly driven by CO and other long-lived species. We

also show that 90% of non-CO annual surface OHR is from NMOGs and that FURA (furan, 2-methylfuran, and 2,5-dimethylfuran) and ethene are important globally for reactivity with phenol more important at a local level in the boreal regions. To our knowledge, this is the first quantification and characterization of the impact and importance of fire for atmospheric reactivity and the first representation of both lumped furans and phenol from fires in GEOS-

Chem. However, our analysis is almost certainly a lower limit on the magnitude of reactivity from fire NMOGs because we do not comprehensively include all species emitted from fires, given that for many of these their global EFs and product formation are not well understood. To further improve the representation of fire NMOGs in models, more measurements of speciated NMOG and total OHR are needed to help constrain both the total emissions and reactivity of

NMOGs, particularly during field campaigns with fire influence. Further development of oxidative chemical mechanisms for highly reactive NMOGs are also needed to ensure that models better capture the exported reactivity from fires. Finally, while we substantially increase the mass and reactivity from fire NMOGs represented in our model, more work is needed to constrain low-volatility NMOGs that are precursors to SOA.


As fires become more intense in the western US and in other temperate and boreal regions due to climate change (e.g., Westerling, 2016; Westerling et al., 2006; Abatzoglou and Williams, 2016; Senande-Rivera et al., 2022) and human forcing leads to different burned area trends globally (Andela et al., 2017), it is becoming ever more important to improve our understanding of fire

emissions, their reactivity, and their impact globally. Our work shows that NMOGs from fires contribute substantially to atmospheric reactivity, both locally and globally, highlighting the urgent need to further constrain the sources and transformations of these species.

Data availability



The GEOS-Chem model is publicly available at: https://zenodo.org/record/4618180 (GEOS-Chem, 2021). The DC3 campaign data are available at https://www. eol.ucar.edu/field_projects/dc3 (last access: February 18, 2018), and the ARCTAS campaign data are available at: https://www-air.larc.nasa.gov/cgi-bin/ArcView/arctas (last access: February 18, 2018). ATTO data are available at https://attodata.org/ (last access: June 4, 2021). FIREX-AQ data are

available at https://www-air.larc.nasa.gov/cgi-bin/ArcView/firexaq (last access: April 16, 2022).

Supplement

See SI at [link added by ACP]

Author contribution

CLH and TSC formulated the research question and wrote the paper with input from JHK. TSC performed the modeling and analysis. ECA, DB, MC, AE, GG, RSH, FP, JP, EYP, NGR, AR, CW, AW, JW, and LX measured CO, VOCs, and OHR during ARCTAS, FIREX-AQ, and ATTO and provided input on the manuscript.


Competing interests

The authors declare that they have no conflict of interest.

Acknowledgements

We thank Kelvin Bates for advice on implementing the aromatic hydrocarbon updates and ethene/ethyne chemistry  that were not yet available in the standard GEOS-Chem model. We also thank Mat Evans for discussing source attribution approaches. We acknowledge Tom Ryerson for making measurements of CO during FIREX-AQ and William Brune for OHR measurements and Teresa Campos for CO measurements during DC3.


Financial Support

This work was supported by the U.S. National Science Foundation (NSF AGS 1936642). This material is based upon work supported by the National Center for Atmospheric Research, which is a major facility sponsored by the National Science Foundation under Cooperative Agreement
No. 1852977. ECA and RSH were also funded in part by NASA Award No. 80NSSC18K0633. JP was supported by the NOAA Cooperative Agreement with CIRES, NA17OAR4320101. The University of Innsbruck PTR-MS measurements were supported by the Austrian Federal Ministry for Transport, Innovation and Technology (bmvit, FFG, ASAP).

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
