# Peer review of "An Improved Representation of Fire Non-Methane Organic Gases (NMOGs) in Models: Emissions to Reactivity"

_Atmospheric Chemistry and Physics, 2022_

## Author Comment (AC1)

RC1:

The authors have pushed forward a critical topic in atmospheric science, a full accounting of carbon emissions from important combustion sources. They have demonstrated successfully that they have made a substantial improvement to the GEOS-Chem model using recent datasets better describing speciated NMOG emission factors from fires. Armed with better emissions and some improved chemical mechanisms, the authors have evaluated the model with available measurements and then used the model to offer new insights into how fire-based NMOG emissions are distributed across the globe, and where their impacts might be felt. The paper is written well, and the study is mostly designed soundly. I have some concerns though about the model development choice to exclude fast-reacting species. I also have some presentation-relevant suggestions that follow.

Thank you for your comments. We have responded in blue to individual comments below.

**Specific Comments:**

1) Lines 284-292: I appreciate the authors including this detailed discussion to reconcile the new emission factor updates with the source datasets and lab measurements at a high level before introducing the CTM complexities. However, I failed to understand some key points. The new model is capturing 72% of the NMOG carbon mass, and I think the reference point for that percentage is the total of the EFs from the same dataset (i.e. Andreae 2019)? If the point of the study is that understanding and accounting for all NMOG is critical, then why purposefully leave any NMOG carbon mass out? Why not lump it with existing model species or add 1 or 2 new lumped species to capture that reactivity? The authors argue that so many of the fast-reacting species can be ignored because they will be oxidized before transporting out of the model domain, but, as the authors also state, their products (which could be longer-lived fragments) could be quite important. If the authors are sure that species won't be transported, then why not modify the chemical mechanism so that this mass can be added as non-transported species but still contribute to oxidation products that already exist in the model? As an aside, can the authors please state the contribution of the 'unspeciated or unidentified NMOG' from Coggon et al. (2019) to give a better idea of how large the gap still may be?

We thank the reviewer for this question. We did consider a "lumped" highly-reactive VOC approach, but this is not feasible given the lack of EFs for these species (i.e. our calculations on carbon mass balance are associated with the western U.S. EFs from Koss et al. and cannot be applied globally). Furthermore, we felt that providing a lumped near-field reactivity source, without describing the oxidative chemistry of these species and hence conserving carbon, would be unphysical. We hope that our work highlights the importance of characterizing the emissions and chemistry of these NMOG species from fires such that additional observational efforts might be deployed to constrain them. We have added the following text to the manuscript:

"A lumped highly-reactive VOC may provide a means of describing this near-field reactivity in the model (though the oxidation products and their reactivity would be poorly described by such an approach). However, given that EFs for these highly reactive species are not globally characterized, there is currently no meaningful way to estimate the emissions of such a lumped VOC."

"We are unable to perform a global NMOG carbon accounting given that many EFs (shown in red in Fig. 1) are only available for a subset of ecosystems (here primarily western U.S. fuels)."

As requested by the reviewer, we have also added that ~25% of calculated primary OH reactivity in Coggon et al. (2019) is unidentified.

2) Lines 292-299: Again, why leave over half the reactivity out of the model if you already have some emission factors to go on, even if they are not global EFs. Is there a discussion somewhere in the manuscript for what is required for an EF to be global? Do you just need data for more tree and grass species? Why not run a version of GEOS-Chem with all of the red species for the simulations targeted to the U.S. measurement campaigns?

This is a global study by design, and EFs vary significantly with ecosystem. The work of Koss et al. measured western fuel types (loosely western-type shrub and grasses and temperate forests). Even within the US, this dataset would not include agricultural EFs, which are important in the southeastern US, and temperate forest and grassland EFs for ecosystems different from the western US.  It would be highly inappropriate to apply western U.S. EFs globally. We add the following text to explain this:

"(we note that these fuel types include shrub, grasses and temperate forests representative of the western US only)"

And modify this sentence:

"while EFs for other fuel types can vary substantially in magnitude, they generally provide a similar relative ranking"

3) Perhaps some of my confusion would be resolved if the authors systematically distinguished between local-scale reactivity, which would be better constrained by Coggon et al. (2019), and downwind (or globally relevant) reactivity, which is what is being improved in this study. If the authors agree, then consider articulating this difference when reporting how much of the reactivity the model can explain in different contexts. Should we expect global models to explain all the reactivity seen in the lab? Are the ATTO cOHR measurements local or at-scale (ties into part of comment 4)?

We note that this work is designed to increase the comprehensiveness of the fire NMOGs in the model. It is not designed to target either near-field or far-field reactivity. The availability of global EFs primarily dictates what we can include in our model simulations. The discussion of

near-field vs far-field is included in Section 3 to provide some context for the species which we cannot include in the model – and this includes both missing near-field reactivity and far-field reactivity from the products of these species. This discussion addresses the reviewers' question regarding what we can expect the global model to represent. We do not agree that it is useful to frame our study as improving only downwind reactivity.

4) Line 420-424 and Fig. 5: The color scheme is confusing. Is the red boxplot in the left panel driven by GFED or FINN emissions? If the former, then please add the FINN boxplot to the left panel. Please also add a supporting figure that shows the contributions of species (or species categories) to the GFED and FINN results so that one can see explicitly why the model update doesn't seem to matter so much. An alternative interpretation could be that both FINN, GFED, and GEOS-Chem are missing emissions of the faster-reacting species, and that these species are playing a significant role in the obs data in addition to the species that are scaled up in the FINN inventory (i.e. maybe those species are scaled up too high). A missing piece of the analysis here is the transport time from the fire to the ATTO site. How much of those fast-reacting species are still going to be around?

Thank you for pointing out this confusion with the legend. The red box in the left panel is driven by GFED4s and thus is consistent across the figure. We have updated the label and added the FINN boxplot to the left panel as well to ensure that this is clear.

A recent paper Pohlker et al. (2018) suggested that some of the smoke reaching the ATTO site is ~2-3 days old, so fast reacting species would no longer be present. However, fires are episodic in nature, and we do not have specific estimates for the lifetime of smoke sampled during the observations that we use. We have, however, added a sentence that fast-reacting species may have reacted away prior to reaching the ATTO site:

"Because the ATTO site is downwind of fires (one estimate for a different fire season suggested smoke was ~2-3 days old when it was measured (Pöhlker et al., 2018)), it is also possible that fast-reacting species are no longer present in the air sampled at ATTO."

5) Could the authors provide a table summarizing the land-use types that are used to parameterize fire EFs in GEOS-Chem. Only three types are highlighted in the SI (Savannah, Temperate Forest, and Agriculture). What about Boreal, Tropical, etc.? It would be better to see emissions for all of the types available in GEOS-Chem. I think this would help readers connect with statements about western US EFs versus global EFs. Fig. 10 is especially hard to interpret without some context about how much variability there is now in speciated EFs across the world.

The GFED4s inventory used in the model includes 6 land-use types. We now list these in the paper when we introduce the inventory. We would refer you to the paper describing GFED4s (van der Werf et al. 2017), for more details on the dry matter and emissions factors (Table 1 in van der Werf) associated with each land type. See also Table 1 in Carter et al. (2020) for EFs by biome type. The EF compilations from Akagi et al. and Andreae are based on fuels measured from regions around the world, such that the "savanna" EF is globally representative. Thus, it would be inappropriate to use western U.S. fuels only to specify these global EFs. We only

provided comparisons in the SI for the two land types where Koss et al. (2018) provided lab measurements (savannah and temperate forest) and for agricultural fires, which is the other important land type in CONUS and which illustrates that Andreae and Akagi are quite similar for other land types.

6) Figure S8 is interesting, and I think there is a lot more to discuss than what is provided so far. The discussion of this figure is limited to pointing out differences between the tropical and boreal case decay-rates. But is that because of latitude, or some kind of marine chemistry at play? Would you find the same fast loss for a boreal fire closer to the ocean? Is the orange trend emissions or is it concentration? Why is there so much scatter in the trends over land? Are these just for surface concentrations? If so, maybe column sums would be more appropriate. How do you determine how the trends follow the fire plume? It would be instructive to see how the species contributions to NMOG reactivity change along this axis, and see what notable differences there are between tropical, temperate and boreal fires. Are there differences when emissions are injected into the free troposphere vs. when they are trapped at lower altitudes? This aspect of the study could be very helpful when planning or interpreting future aircraft campaigns.

We are glad that the reviewer finds Fig. S8 interesting. This simple calculation was meant to provide an illustration of thinking about OHR dissipation in two large regions where fire plumes could be somewhat resolved. It is however, challenging to extract and attribute quantitative information given the variability in transport timescales and spatial patterns, hence our use of the words "suggest" and "likely" in our manuscript and our relegation of this figure to SI. We have added a sentence to the manuscript:

> "Further exploration within a Lagrangian framework may provide more insight into the evolution of OHR downwind of fires."

In response to the reviewer's specific questions: The orange trend are emissions (described in the caption), so there is a lot of scatter over land as we cross over different fires. Because most of the dissipation happens over land (either in boreal regions or larger drop-offs seen in the African examples before the land break, the differences are likely driven by latitude, and thus OH concentrations, rather than marine chemistry. In this study, all emissions are injected into the boundary layer (and we show surface OHR), so it is not possible to look at the question of injection height.

**Minor Comments/Typos:**

Line 34: semicolon should be a comma.

We have made this change in text.

Line 47: Reads like ATTO was in the US along with FIREX-AQ and DC3. Recommend rewriting.

We have rewritten for clarity.

Line 51: exceeding à greater than

We have made this change.

Line 68: particulate matter with diameter smaller than 2.5 microns

We have accepted his change.

Lines 128-130: recommend moving 'from fires' to line 130: "…and measured OHR from fires and that furans…"

We have implemented this change.

Line 198: "multiplied by its concentration as follows:"

We have implemented this change.

Fig. S4: It would help to have a table that defines all of the chemical species names here. Some names are more ambiguous than others, but I think readers who aren't steeped in chemical mechanism modeling would find a key useful.

We have added a table in the SI to define the chemical species.

Fig. S4: Why are furans not in these figures? Can they be added? Or if they are already present, which species do they fall under?

We have now added furan EFs to Fig. S4.

Fig. 1a: This is a nice figure! While the gray box as defined is sort of interesting from a model development stand-point, particularly for folks used to the GEOS-Chem or global model context, I think there could be a higher-impact option. Since the take-home point (I think) is to separate fast-reacting from slow-reacting species, can you instead add regions illustrating the timescales for local, regional, continental, and hemispheric transport? Pretty much all readers should have an intuitive grasp of this, but I think it would be useful to picture it directly behind the quantified OH lifetimes.

Thank you for this comment. We retain the GEOS-Chem related box because it is a figure constructed around how species treated in GEOS-Chem, so the GEOS-Chem timescale is relevant. However, we have now added these timescales to the text for the reader's reference:

> "The timescales for regional (~7 days), continental (~18 days), and hemispheric (~1 year) transport may also be relevant."

Fig. 1a: Isoprene is blue. Does this mean that the standard model didn't have it? I'm confused about whether blue means that it is a brand-new model species, or whether it is just now getting fire emissions for the first time.

Blue means that it was not in the model as a fire emission, and so we have used Andreae EFs to add it. We have added some text to the caption to clarify this.

Fig. 1b: The standard definition of the VOC/IVOC cutoff is halfway between 1e6 and 1e7 in log space, although I agree that Ahern et al. (2019) went to 1e6, as you write. Consider moving the guiding line. Also please add 298 K (I assume) to the y axis label.

The exact choice of the VOC/IVOC cutoff is arbitrary and doesn't affect the present analysis. The difference between the cutoff we used and the one the reviewer suggests is only a factor of 3.2, the change in volatility associated a single CH2 group. To be consistent with Ahern 2019 we prefer to keep the 1e6 ug/m3 value.

We have added 298 K to the y axis as requested.

Line 274: I am not yet convinced by Fig. 1 that transport is not important for all the red species. I am missing the OH concentration assumed for this axis. How consistent is that with what is seen in wildfire plumes?

Thank you for pointing out this omission. We use 1E6 molec cm$^{-3}$ as our [OH] concentration, which we have now clarified in the caption. The surface [OH] in the model varies globally by a factor of ~5. Liao et al. (2021) suggest that plume-average concentrations range from 0.5-5.3 E6 molec cm-3, suggesting that our assumed value is also within a factor of 5 of in-plume [OH]. We add the following to the text:

" We use an assumed OH concentration of $1x\ 10^6$ molec cm$^{-3}$; local values likely are within a factor of 5 of this value, given the simulated variability and estimated plume-average OH concentrations in fire plumes (Liao et al., 2021)."

Line 296: Think more about how to refer to the "minor species". They don't sound minor based on their contribution to reactivity. "minor species by mass" or just "fast-reacting species" (?)

We have changed to "fast-reacting" in text.

Line 298: "all of these fractions" is vague. I think the NMOG% are relative to the EF datasets and the OH reactivity percentages are relative to Coggon et al. (2019), no?

We have clarified in text.

Fig. 9: I recommend reconceptualizing this figure into a format that can be better-used as a reference. Pie charts are notoriously difficult for humans to interpret, and the large number of colors, and varying sizes of the circles make that even tougher. What about a three-panel figure of bar charts? Panel 1 could be stacked bars showing the contribution of fire and non-fire cOHR over the globe and land. Panel 2 could be fractional stacked bars showing the break-down of CO, NO2, Other, and total NMOG for both cases. Panel 3 could then be fractional stacked bars showing the contribution of each NMOG species to total NMOG. There are a lot of species, so maybe this panel could stretch horizontally across the bottom.

The objective of this figure is to generally illustrate the relative importance of species contributing to fire OHR. The absolute quantities are less essential, particularly given that we are showing large-scale averages (global or over land) for a quantity that varies substantially spatially. While we therefore agree with the reviewer in certain contexts regarding the superiority of bar plots over pie charts, in this case, the pie chart illustrates the information that we intend to convey more simply than the proposed change.

Fig. 10: Do you have the data from the standard model configuration to depict before and after for each of these regions? I recommend adding it if you do. Second, another dataset to consider adding is a measure of acres burned for each of these regions. This way you can depict both the potency of fires in each region as you are now and the magnitude of the signal. For example, I wonder about the abundance of reactivity exported from Asia versus Alaska and how much of the data here is adjusted low by the huge land area of Asia (I assume these data are averaged in space and time?). Finally, please grab a new color-scale. I'm not colorblind, and the purples somewhat ran together for me. Mixing the green with these reddish colors might also be problematic, but I'm not sure.

We have added a "before" version from the standard model to the SI. We have also added average burned area quantities for each region to Figure 10 in response to the comment and inserted a sentence in text about the potency of fires in each region and the magnitude of the cOHR signal.

We appreciate the concern about the colors and have confirmed with a color-blind testing application that these are accessible.

Line 548: recommend changing generally to potentially. Unless I'm mistaken, the model still hasn't been evaluated successfully in Africa for cOHR.

We have integrated this suggestion in text.

Line 566-567: Recommend rephrasing the comment about phenol. The model results predict phenol is important just in the boreal regions, but it also missed the phenol measurements during FIREX-AQ. I agree that it's hard to say how much of that might be non-fire anthropogenic sources, but I encourage you not to discount phenol in the mid-latitudes or even tropics just yet. There could also be secondary production that the mechanism is missing.

We have integrated this comment in text.

RC2:

Carter et al. present an analysis in which they update the treatment of fire VOCs in the GEOS-Chem CTM to account for recent emission studies. Along with EF updates they incorporate recent expansions of the model chemical mechanism, and add fire emissions for relevant species where that source was previously missing. They compare the model output to atmospheric observations and draw some conclusions regarding the importance of fires for global emissions of reactive carbon and reactivity, and point to some areas needing future work.

Overall the paper is straightforward and relevant to ACP. I recommend publication with a few minor comments/suggestions below.

Thank you. Our responses to each comment are below in blue.

254-256 and associated discussion of the percentage mass and reactivity captured by the model. If I understand correctly, with the updates you now have ~40TgC of fire emissions in the model, and you estimate that this is capturing 72% of the total mass emissions and 45% of the emitted reactivity. But earlier it was indicated that fires emit > 400Tg/y, based on Akagi. Please discuss the reasons for and implications of this disparity. The paper does include some discussion and caveats about the estimates being a likely lower limit, but that discussion does not appear to cover a potential disparity of this magnitude.

Thank you for raising this point. We agree that the ~40 Tg C is lower than the estimate from a variety of papers of 100-200 Tg C, which we have now revised. Table 5 in Akagi et al. clarifies that the larger 400 Tg number that we previously cited includes several categories that we are not representing in our biomass burning estimate, including biofuel, open cooking, and garbage burning that we have now removed from their estimate to get 200 Tg. Andreae and Merlet 2001 estimates that ~100 Tg C of fire NMOG are emitted. We have added a sentence that we are still underestimating these literature values, potentially because smoke inventories underestimate small fires:

> "We note that the global total of speciated NMOG emissions estimated by GFED4s (27 to 48 Tg C yr$^{-1}$) are smaller than the simple calculations in the literature (100-200 Tg) (Akagi et al., 2011; Andreae and Merlet, 2001) likely because we are not representing all possible species and because GFED4s is known to underestimate emissions from small fires (van der Werf et al., 2017; Randerson et al., 2012)."

We would also like to clarify that the percentages that you refer to are part of our very simple calculation looking at only two fuel types for which we have EFs from Koss et al. (savanna and temperate forests). These percentages should not be directly compared to the total NMOG emitted. We have clarified that in text.

250, "In the model, the oxidation of FURA with OH produces butenedial since that has been shown experimentally with an estimated carbon balance of 100% C". This is true for furan but not for the species lumped into FURA.

The reviewer is correct. We have clarified the text in the manuscript:

> "In the model, the oxidation of FURA with OH produces butenedial following Bierbarch et al., (1995) who show that furan forms butenedial with an estimated carbon balance of 100% C"

The writing is in generally very well-done. In places awkward or excessive use of puncutation (e.g. commas, nested parentheses) renders things less clear than they could be. The authors may wish to revisit this

Thank you. We have edited for this.

307, things get a bit awkward here with the use of FIREX vs FIREX-AQ. Earlier FIREX was used to refer to the lab component but here it appears to refer to the airborne campaign. And later in the section you spell out the FIREX-AQ acronym when it has already appeared numerous times before that. Likewise ARCTAS is not spelled out at first use but rather on line 390 which makes things confusing

Thank you for pointing out these discrepancies. We have corrected this throughout the text.

103, "greater variability". Unclear, greater than what?

We have clarified in text.

105, extra parenthesis

We have deleted it.

153, please indicate time resolution of the boundary conditions.

We have added this information.

197, I think you mean "product" rather than "sum" here

We have clarified in text.

221-222 and Fig 1 caption, please state the assumed OH concentration that is implicit in the lifetimes provided

We have added this information.

Fig 1, the use of an outer black circle to designate species whose oxidation products are known is not ideal since some of the symbols are solid black

We have changed to grey in text.

244, placement of "(RCHO)" in this sentence makes it seem as though it is referring to furfural when I believe it is actually referring to "aldehydes with three or more carbon atoms"

We have fixed this in text.

295, there appears to be an inconsistency between the values given here versus the abstract, 45% / 49% for reactivity?

Thank you. We have corrected that in text.